# Temporal Mortality Trends Attributable to Stroke in South Asia: An Age–Period–Cohort Analysis

**DOI:** 10.3390/healthcare12181809

**Published:** 2024-09-10

**Authors:** Ruhai Bai, Minmin Li, Ashok Bhurtyal, Wenxuan Zhu, Wanyue Dong, Di Dong, Jing Sun, Yanfang Su, Yan Li

**Affiliations:** 1School of Public Affairs, Nanjing University of Science and Technology, Nanjing 210094, China; ruhaibai@hotmail.com; 2Clinical Medical Research Center, Children’s Hospital of Nanjing Medical University, Nanjing 210008, China; 3Department of Nutrition and Food Risk Monitoring, Shaanxi Provincial Center for Disease Control and Prevention, Xi’an 710054, China; lmm123@stu.xjtu.edu.cn; 4Central Department of Public Health, Institute of Medicine, Tribhuvan University, Kathmandu 46000, Nepal; ashokbhurtyal@iom.edu.np; 5Health Science Center, Xi’an Jiaotong University, Xi’an 710061, China; wenxuanzhu828@163.com; 6School of Health Economics and Management, Nanjing Chinese Medicine University, Nanjing 210023, China; wanyuedong@njucm.edu.cn; 7Duke Global Health Institute, Duke University, Durham, NC 27710, USA; dd154@duke.edu; 8Global Health Research Center, Duke-Kunshan University, Kunshan 215316, China; 9Rural Health Research Institute, Charles Sturt University, Leeds Parade, Orange, NSW 2800, Australia; jinsun@csu.edu.au; 10Department of Global Health, University of Washington, Seattle, WA 98105, USA; yfsu@uw.edu; 11School of Public Health, Shanghai Jiao Tong University School of Medicine, Shanghai 200025, China

**Keywords:** stroke, brain infarction, cardiovascular disease, cerebrovascular disease, South Asia

## Abstract

South Asia contributes the most to stroke mortality worldwide. This study aimed to determine the long-term trends in stroke mortality across four South Asian countries and its associations with age, period, and birth cohort. In 2019, nearly one million stroke deaths occurred across South Asia, and the associated age-standardized mortality rate (ASMR) was 80.2 per 100,000. Between 1990 and 2019, India had the largest decrease in the ASMR (−35.8%) across the four South Asian countries. While Pakistan had the smallest decrease in the ASMR (−7.6%), an increase was detected among males aged 15 to 34 years and females aged 15 to 19 years. Despite a 22.8% decrease in the ASMR, Bangladesh had the highest ASMR across the four South Asian countries. Nepal reported a witness increase in the stroke ASMR after 2006. Improved period and cohort effects on stroke mortality were generally indicated across the analyzed countries, except for recent-period effects in males from Nepal and cohort effects from those born after the 1970s in Pakistan. Stroke mortality has decreased in the four South Asian countries over the past 30 years, but potentially unfavorable period and cohort effects have emerged in males in Nepal and both sexes in Pakistan. Governmental and societal efforts are needed to maintain decreasing trends in stroke mortality.

## 1. Introduction

A stroke occurs when the blood flow to an area of the brain is interrupted, resulting in some degree of permanent neurological damage. Strokes are classified into two major categories: ischemic (occurs when a blood vessel supplying blood to the brain is obstructed) and hemorrhagic (occurs when a weakened blood vessel ruptures). For ischemic stroke, five subtypes are classified by the TOAST (large-artery atherosclerosis, cardioembolism, small-vessel occlusion, stroke of other determined etiologies, and stroke of undetermined etiology). Hemorrhagic stroke can be classified into two subtypes: intracerebral hemorrhage (bleeding into the brain parenchyma) and subarachnoid hemorrhage (bleeding into the subarachnoid space) [1].

Stroke is a major contributor to disability and death worldwide [1,2]. If the current trends continue, there are expected to be 200 million stroke survivors, and each year thereafter, 12 million stroke deaths are expected to occur by 2050 [3]. South Asia is the most populous region worldwide, and its situation is even worse since the region accounts for more than 40% of stroke deaths worldwide [4]. Furthermore, the incidence of stroke is higher in South Asians [5,6].

In 2013, the United Nations declared the aim of reducing the relative risk of cardiovascular mortality by 25% by 2025 [7]. However, national healthcare authorities and health policymakers in South Asia consider noncommunicable diseases to be a low priority [4]. Global campaigns addressing stroke incidence and mortality will not achieve the desired reductions without considerable success in this region [4]. Therefore, an understanding of recent temporal trends in stroke deaths across South Asian countries remains imperative for assessing the impact of current interventions and guiding future policies.

Available stroke mortality data from South Asia are mostly confined to a few countries and specific time periods [5,8,9,10,11]. However, few studies have explored changes in stroke mortality over time or among different age groups in South Asia. The potential effects of age, period, and cohort on stroke mortality in South Asia are also currently unknown. To address these gaps, we analyzed trends in stroke mortality and the impacts of age, period, and birth cohort effects by sex across the four most populous South Asian countries over the last 30 years. Our results cover the available body of scientific knowledge from the most regularly updated records for this region. They also provide important insights into prioritized resource allocation, with the aim of preventing stroke among susceptible populations.

## 2. Materials and Methods

### 2.1. Data Sources

We extracted data from the Global Burden of Disease Study (GBD) 2019, which provides a comprehensive assessment of age- and sex-specific mortality for 369 causes of death in 204 countries and territories from 1990 to 2019 [12]. The GBD 2019 data are available for use in research and analysis at http://ghdx.healthdata.org/gbd-results-tool (accessed on 23 May 2021). The GBD 2019 uses anonymized and aggregated data, for which the Institutional Review Board of the University of Washington provided an ethical review waiver.

The four South Asian countries selected for this analysis were Bangladesh, India, Nepal, and Pakistan. Details on the original data of each country are described in the GBD 2019 flagship paper [12]. In this study, stroke was defined by the World Health Organization clinical criteria as rapidly developing clinical signs of (usually focal) disturbance of cerebral function lasting more than 24 h or leading to death. The 9th and 10th revisions of the International Classification of Diseases were used to define stroke (ICD10 codes G45–G46.8, I60–I63.9, I65–I66.9, I67.0–I67.3, I67.5–I67.6, I68.1, I68.2, I69.0–I69.3; ICD9 codes 430–435.9, 437.0–437.2, 437.5–437.8) [12]. To estimate the mortality attributed to stroke, the Cause of Death Ensemble modeling (CODEm) framework was used for the vital registration and verbal autopsy data. CODEm is a flexible modeling tool to produce estimates of death for all locations [12]. Temporal stroke trends were age-standardized to the GBD 2019 global age-standardized population.

National census data from BRICS countries were used to estimate populations. An improved Bayesian hierarchical model was used to estimate age-specific populations to ensure maximum internal consistency. The details of this process were described in a previous study [13].

### 2.2. Statistical Analysis

An age–period–cohort (APC) model was used to assess the effects of age, period, and cohort on stroke mortality. In this model, the age effect represents how the risks of different outcomes vary between age groups, the period effect represents changes in results over time and affects all age groups simultaneously, and the cohort effect represents the differences in results between groups with different birth years [14]. The APC framework provides additional insight into understanding time-varying elements in epidemiology [15,16,17] and is particularly suitable for the current analysis. Specifically, the following parameter was estimated in this study [18]: net drift, which indicates the annual percentage change in the expected age-adjusted stroke mortality rates over time. Local drift indicates annual percentage changes in stroke mortality rates for each age group during the study period. The longitudinal age curve, which represents the age effect, indicates the expected age-specific stroke mortality rate in a reference cohort adjusted for period effects. The period rate ratio (RR), which represents the period effect, indicates the stroke mortality rate in a period versus the reference one. The cohort RR represents the cohort effect, which indicates the TBL mortality rate in a cohort versus the reference cohort [18].

To conduct the APC analysis, mortality and population data were arranged into consecutive 5-year periods from 1990 to 2019 and successive 5-year age intervals from 15–19 years to 90–95 years. The sample consisted of 20 successive cohorts from those born from 1905–1909 to 2000–2004, with the birth cohort from 1950–1954 being the reference cohort. The APC Web Tool (Biostatistics Branch, US National Cancer Institute) was used to estimate all the parameters [19]. Estimable functions were determined via Wald χ^2^ tests. These estimates comply with the GATHER statement. All the statistical tests were two-tailed, and *p* < 0.05 was considered significant.

## 3. Results

### 3.1. Stroke Mortality Trends

The changes in death rates attributable to stroke across the selected South Asian countries from 1990 to 2019 are presented in Table 1 and Figure 1. In 2019, nearly 1,000,000 stroke deaths occurred across South Asia, and the age-standardized mortality rate (ASMR) for stroke was 80.2 per 100,000. Between 1990 and 2019, despite the increase in total stroke deaths and in the relative proportion of all-cause deaths attributable to stroke, which increased from 523,900 to 978,900 (+86.8%) and from 4.8% to 8.2% (+70.8%), respectively, the ASMR decreased from 119.0 to 80.2 per 100,000 (−32.6%).

Over the past 30 years, stroke ASMRs in Bangladesh and Pakistan have been higher than the regional average. Despite a 22.8% overall decrease in the stroke ASMR, Bangladesh still had the highest ASMR and relative proportion of all-cause deaths attributable to stroke across the four South Asian countries. Stroke mortality in Bangladesh continued to increase until 2007 and then decreased. Pakistan had the smallest decrease in the ASMR due to stroke, from 124.3 per 100,000 in 1990 to 114.9 per 100,000 in 2019 (−7.6%). Between 1990 and 2019, although India experienced a steady increase in stroke deaths, rising from 0.4 million to 0.7 million (+85%), it achieved the largest ASMR reduction (−35.8%) among the four South Asian countries. Since 2010, India’s ASMR has been lower than Nepal’s, making it the country with the lowest mortality among the studied nations. Nepal had the lowest ASMR (107.6 per 100,000) and relative proportion of all-cause deaths attributable to stroke (3.8%) in 1990 and exhibited an overall decrease in the ASMR from 1990 to 2019. However, after 2006, its stroke-attributed ASMR increased, putting it above the average level of South Asians by 2019 (80.4 vs. 80.2 per 100,000). After 2008, the relative proportion of all-cause deaths resulting from stroke in Nepal also exceeded that in Pakistan and India.

### 3.2. Age-Specific Mortality Rates for Stroke

We arranged the mortality and population data into consecutive 5-year groups from 1990–1994 (median 1992) to 2015–2019 (median 2017) and 20 consecutive cohorts from 1905–1909 (median 1907) to 2000–2004 (median 2002). Figure 2 and Appendix A show the trends in the stroke mortality rates across the four South Asian countries, while Appendix A lists the number of stroke deaths across each age group. Figure 2A–D depict a decreasing trend in stroke mortality between 1990–1994 and 2014–2019. As shown in Figure 2E–H, Bangladesh and Pakistan first exhibited an increase and then a steady decrease in stroke mortality across all age groups. Stroke mortality has fluctuated across birth cohorts in India, whereas Nepal has experienced a decrease and then an increase from early to late birth cohorts.

### 3.3. Net and Local Drifts between Age Groups

As shown in Figure 3, the general net drift in stroke mortality substantially decreased during the study period in Nepal (−1.7 (95% CI, −1.9 to −1.6)), India (−1.5 (95% CI, −1.8 to −1.2)), Bangladesh (−1.1 (95% CI, −1.5 to −0.8)), and Pakistan (−0.5 (95% CI, −0.5 to −0.4)). There were marked differences in the overall annual change between sexes in Nepal (−1.0 (95% CI, −1.3 to −0.8) for males and −2.6 (95% CI, −2.9 to −2.3) for females), with a smaller improvement in mortality in males than in females.

Local drift values were predominantly less than 0 for most age groups of both sexes, indicating a decrease in stroke mortality. The exceptions were males aged 15–34 years in Pakistan (annual increase in mortality ranging from 0.4 (95% CI, 0.1 to 0.7) to 1.2 (95% CI, 0.6 to 1.8)) and females aged 15–19 years in Pakistan (annual increase in mortality of 0.6 (95% CI, 0.0 to 1.1)). The largest improvements were for Indian females aged 15–19 years (−3.9 (95% CI, −7.2 to −0.5)) and Indian males aged 15–19 years (−3.2 (95% CI, −6.0 to −0.3)). Among elderly individuals, India had the fastest decrease in mortality across these four South Asian countries.

### 3.4. APC Effects on Stroke Mortality

Figure 4 shows estimates of the effects of age, period, and cohort on stroke mortality while adjusting for the other two factors. Figure 4A–C show the long longitudinal age curve of the mortality of stroke, which was the expected mortality rate by age group in the reference population adjusted for period effects. The mortality rate increased exponentially with age in all four South Asian countries (Appendix A). Bangladesh has the fastest increase in stroke mortality with age.

Figure 4D–F shows the estimated period effect. Favorable period effect trends were found in most South Asian countries, where the period RR decreased in the last ten to twenty years. The exception was males in Nepal, where period RR increased in the last fifteen years.

Figure 4G–I shows the estimated cohort effect. The most striking improvements across birth cohorts were observed in India, with a general progressive improvement in stroke mortality in both males and females. Continuous improvements in stroke mortality across birth cohorts have been reported in older cohorts in Pakistan, but these improvements appear to have stagnated and deteriorated for Pakistani cohorts born after the 1970s. In Bangladesh, there was an indication of more favorable trends for people born after 1920. In Nepal, the cohort effect has continuously decreased since the 1915 birth cohort, but this favorable trend tended to diminish for males born after 1985.

## 4. Discussion

This study assessed the long-term trends in stroke mortality in four South Asian countries to examine age-, period-, and cohort-specific effects via the APC model. Our results indicate that there have been improvements in stroke mortality across the four South Asian countries over the last 30 years; however, the period and cohort effects indicate potentially unfavorable trends in recent periods (males in Nepal) and birth cohorts (males in Nepal and both sexes in Pakistan).

There were striking differences between the four South Asian countries in terms of stroke mortality at given points and across time periods. Bangladesh consistently had the highest stroke mortality. Meanwhile, India, with stroke mortality lower than Nepal’s after 2010, became the country with the lowest mortality among the four studied nations, which was less than half of Bangladesh’s mortality in 2019. Although stroke-attributed mortality generally plummeted in Nepal, it steadily increased after 2006.

Bangladesh had the highest ASMR among the four selected South Asian countries over the last 30 years. Proximal determinants accounting for its high stroke mortality include an epidemiological transition from communicable to noncommunicable disease, which has been underway for the last few decades [20]; a sharp increase in the prevalence of high blood pressure; and restricted availability and utilization of hypertension treatment [21]. Our analysis indicated that the increasing trend in stroke mortality in Bangladesh reversed in 2007 and that the period effects also decreased after 2009. This may be due to the Health, Population and Nutrition Sector Development Programme, which was established in 2011 to prevent and control noncommunicable diseases [22]. In addition, nongovernmental organizations have also contributed to primary stroke prevention strategies and long-term stroke care measures [4,23]. However, the provision of appropriate healthcare interventions to reduce risk and offer treatment remains a persistent issue in Bangladesh. Furthermore, cigarette smoking, which induces a particular risk for stroke occurrence, has become more prevalent because of the easy availability of tobacco despite the much-hyped control law, thus contributing to increasing consumption patterns across population groups [24].

India has exhibited a continuous improvement in stroke mortality over the last 30 years. This finding is consistent with an earlier report that the ASMRs at all ages among Indian males and females decreased from 2000 to 2015 [25]. The temporal trends in stroke mortality were favorable, with improvements in both the time and birth cohorts. These increases resulted from enhanced medical technologies, public health initiatives in stroke prevention, and improved healthcare coverage. India has also achieved success in tobacco control, which is strongly related to stroke mortality. The prevalence of smoking tobacco of any type in India decreased from 19.8% in 1987 to 8.6% in 2016; the prevalence among males ranged between 36% and 16%, while it never exceeded 3% among females [26]. Despite these achievements, other risk factors, such as poor diet, including high sodium, trans-fatty acid, and sugar-sweetened beverage consumption, and an increasing prevalence of hypertension and diabetes exist in India [27], suggesting that there is further room for improvement.

Nepal significantly decreased stroke mortality before 2006. This trend may be partly attributed to the effective efforts of the Nepal government in addressing rural physician shortages [28]. In Nepal, 83% of the population lives in rural areas [28]. However, 85% of specialists and 56% of public sector doctors work in the central region [28]. Considerable efforts have been made since 1991 to increase access to doctors for the rural population, including encouraging graduates to work in rural areas, increasing preservice education quality, and scaling up the provision of healthcare in the private sector [28]. Policies regarding the prevention and control of tobacco consumption in Nepal introduced in the 1990s also somewhat contributed to reducing stroke mortality [29]. Between 2006 and 2016, the prevalence of male smokers aged 15–49 years decreased from 32.5% to 27.2%, whereas for females, it decreased from 19.6% to 8.4% [29]. However, the rapidly increasing burden of noncommunicable diseases [30] seems to outweigh these efforts; after 2006, stroke mortality began to continually increase in Nepal. Furthermore, the recent period and cohort effects also indicate an unfavorable trend, which sounds an alarm for Nepal and indicates that more effort is needed to effectively control the increase in stroke-attributed mortality.

Stroke mortality in Pakistan has been decreasing over the past 30 years, which reflects some progress in stroke control there. More than 20 years ago, the Pakistan Stroke Society was established to improve stroke prevention and care in Pakistan by raising awareness among both the general public and doctors [31]. Since 2010, a comprehensive and clear strategy for stroke patient care has been provided by the Pakistan Society of Neurology [31]. However, the burden of stroke risk factors has been increasing in Pakistan. Hypertension, an important risk factor for stroke, increased from 19.55% from 1990–1999 to 29.95% from 2010–2017 [32]. Pakistan is also one of the countries with the highest levels of tobacco consumption in South Asia [33]. Although malnutrition still exists, the burden of obesity continues to increase in Pakistan [34]. Moreover, Pakistanis have insufficient awareness of stroke. A survey of the Pakistani general population indicated that only 51% of the respondents identified the brain as the organ affected by stroke [35], approximately 13% of respondents did not know of any stroke risk factors, and 11.6% had no knowledge of its signs or symptoms. An increase in stroke mortality has recently been observed among young people, and the potential cohort effect also indicates an increased risk of stroke in more recent birth cohorts. If these current adverse trends are not effectively controlled, the achievements in controlling stroke in Pakistan are likely to be offset by the increase in risk factors for stroke as birth cohorts progress, possibly resulting in stroke having an even greater burden in Pakistan.

South Asian countries are facing numerous challenges related to stroke, and there is an urgent need to develop cost-effective strategies and interventions to control the stroke epidemic. For a certain country, understanding the risk factors and developing targeted measures are essential for developing effective stroke strategies in the region [4]. The effective diagnosis and control of hypertension, antiplatelet therapy, and smoking control (including the use of chewing tobacco) may be among the most critical intervention areas [4].

The present study had some notable limitations. First, our APC model analysis was based on GBD 2019 data, which include 30 years of cross-sectional data from 1990, and was not a cohort study. Thus, large-scale cohort studies in different countries are needed to confirm the time-specific RRs. Second, the ICD definition of stroke changed (from the ICD-9 to the ICD-10) during the study period according to WHO recommendations. This transition may influence the statistical accuracy of stroke mortality data. However, studies on cardiocerebrovascular disease in the United States and China have suggested that the ICD transformation has minimal effects on stroke estimates [36,37]. Third, while the GBD 2019 implemented numerous measures to increase data comparability, including corrections for underreporting, incompleteness, misclassification, and redistribution of garbage codes, some bias may still have been present. Fourth, in this study, we did not examine long-term trends in stroke attributed to certain risk factors, which could provide more useful information for stroke prevention. Therefore, future in-depth studies are needed to explore these relationships. 

## 5. Conclusions

This study revealed that stroke mortality in the four South Asian countries has generally decreased over the past 30 years. However, potentially unfavorable period and cohort effects have emerged in Nepal and Pakistan. With increasing life expectancy, aging populations, declining infectious disease mortality, and rising stroke risk factors in transitioning economies, stroke may have a significant impact on some South Asian countries in the future. More governmental and societal efforts are needed to prevent a reversal of the downward trends in stroke mortality.

## Figures and Tables

**Figure 1 healthcare-12-01809-f001:**
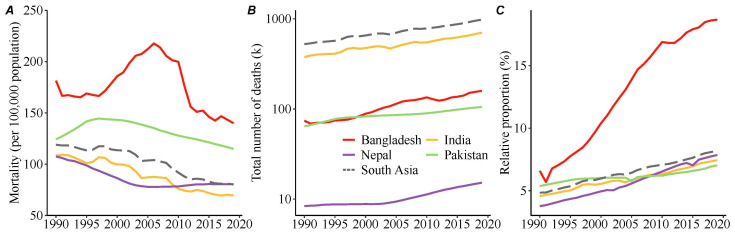
Temporal trends in stroke mortality in South Asian countries between 1990 and 2019. (**A**) Age-standardized mortality rates of stroke; (**B**) the total number of deaths resulting from stroke; (**C**) the relative proportion of stroke to all deaths. The colored lines indicate the relevant value in each South Asian country, and the gray dotted line indicates the overall value in South Asia.

**Figure 2 healthcare-12-01809-f002:**
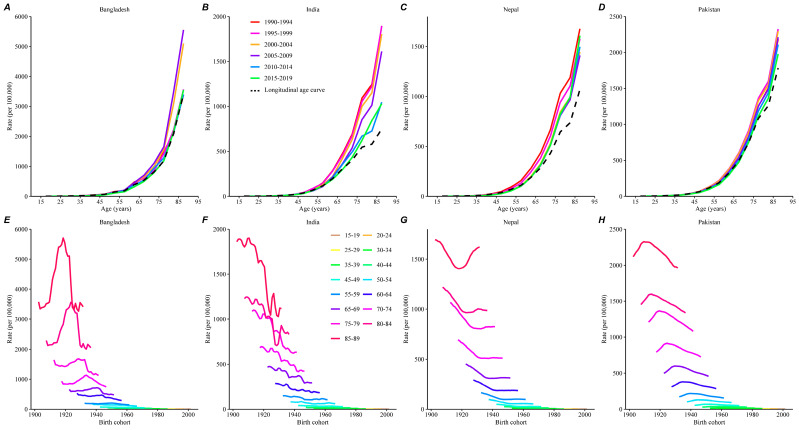
Age-specific mortality rates of stroke by period and cohort across South Asian countries between 1990 and 2019. (**A**–**D**) Survey years were arranged in consecutive 5-year periods: 1990–1994, 1995–1999, 2000–2004, 2005–2009, 2010–2014, and 2015–2019. (**A**–**D**) Stroke mortality rates increased with age. The trend in stroke mortality between the 1990–1994 and 2015–2019 time periods is also shown. (**E**–**H**) The stroke mortality data were arranged into 16 consecutive 5-year age intervals from 15–19 years (median, 17 years) to 85–89 years (median, 87 years). (**E**–**H**) show that India had a decreasing trend in stroke mortality rates across birth cohorts, whereas Bangladesh and Pakistan had initial rising trends toward reductions in all age groups, and Nepal showed a decreasing and then rising trend in all age groups.

**Figure 3 healthcare-12-01809-f003:**
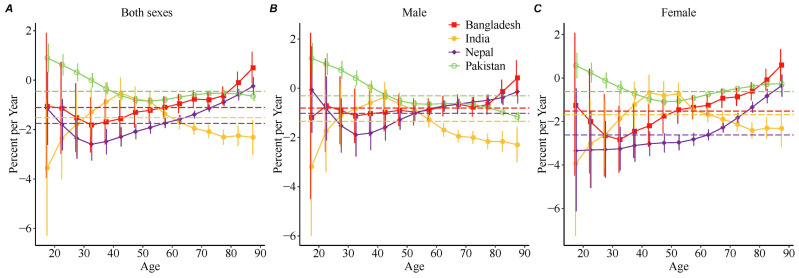
Local drift with net drift values for stroke mortality in South Asian countries from 1990 to 2019. The dashed lines indicate the net drift in each country. Net drift represents the overall annual percentage change, and the values were all <0, indicating substantial reductions in stroke mortality across the study period. The continuous solid line indicates local drift in each age group. Local drift values represent the annual percentage change in each age group. (**A**) both sexes. (**B**) male. (**C**) female.

**Figure 4 healthcare-12-01809-f004:**
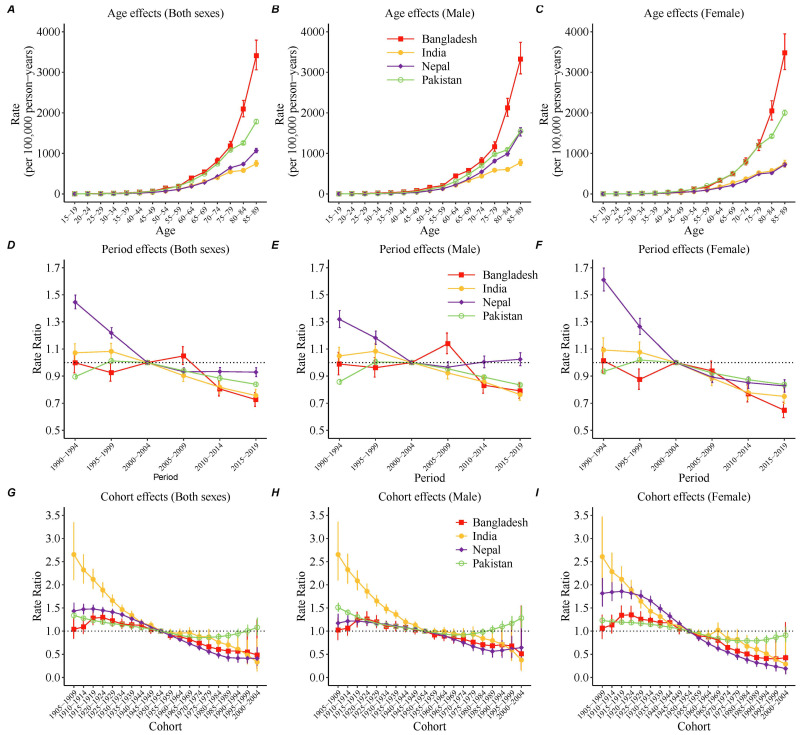
Estimated effects of age, period, and cohort on stroke mortality rates in South Asian countries. (**A**–**C**) Expected age-specific stroke mortality rates (deaths per 100,000 person-years) in the reference cohort (cohort born from 1950–1954), adjusted for period effects, i.e., longitudinal age curves. (**D**–**F**) Period rate ratios and risk of stroke mortality in each period relative to the reference period (2000–2004), adjusted for age and nonlinear cohort effects. (**G**–**I**) Cohort rate ratios and risk of stroke mortality in each cohort relative to the reference cohort (cohort born from 1950–1954), adjusted for age and nonlinear period effects. The error bars indicate 95% CIs.

**Table 1 healthcare-12-01809-t001:** Characteristics of stroke deaths in selected South Asian countries between 1990 and 2019.

	Population				Stroke					
	Total, n × 1000,000		Percentage of Global Population (%)		ASMR ^1^, per 100,000		Deaths, n × 1000		Relative Proportion, %	
	1990	2019	1990	2019	1990	2019	1990	2019	1990	2019
Bangladesh	109.1 (101.3, 117.1)	159.3 (141.2, 177.9)	2.0	2.1	181.4 (153.9, 206.7)	140 (107.0, 169.5)	74.0 (63.3, 83.9)	158.8 (121.9, 193.5)	6.6 (5.7, 7.4)	18.7 (14.7, 20.5)
India	855.6 (791.7, 918.9)	1390.7 (1237.8, 1237.8)	16.0	18.0	108.3 (94.6, 124.8)	69.5 (59.3, 80.1)	377.1 (332.4, 428.4)	699.1 (593.8, 806.6)	4.6 (4.0, 5.2)	7.4 (6.8, 8.1)
Nepal	19.5 (18.0, 21.0)	30.4 (26.6, 34.2)	0.4	0.4	107.6 (80.4, 144.0)	80.4 (63.4, 98.6)	8.4 (6.3, 11.3)	15.2 (12.0, 18.7)	3.8 (2.9, 4.9)	7.9 (6.7, 9.3)
Pakistan	112.8 (100.3, 125.2)	224.1 (207.1, 241.7)	2.1	2.9	124.3 (96.1, 151.9)	114.9 (98.7, 139.2)	64.1 (50.5, 77.8)	105.4 (90.1, 126.6)	5.4 (4.2, 6.5)	7.0 (6.3, 8.0)
South Asia	1097.6 (1032.7, 1164.8)	1805.2 (1650.4, 1971.0)	20.5	23.3	119.0 (103.6, 137.0)	80.2 (70.9, 89.7)	523.9 (462.9, 595.5)	978.9 (864.8, 1095.4)	4.8 (4.3, 5.5)	8.2 (7.6, 8.8)

^1^ ASMR, age-standardized mortality rate.

## Data Availability

The dataset supporting the conclusions of this article is available in the Global Burden of Disease Study 2019 (GBD 2019). Hyperlink to the dataset is http://ghdx.healthdata.org/gbd-2019 (accessed on 12 November 2020).

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
