# Peer review of "Temporal Mortality Trends Attributable to Stroke in South Asia: An Age–Period–Cohort Analysis"

_healthcare, 2024, doi:10.3390/healthcare12181809_

Round 1

Reviewer 1 Report

Comments and Suggestions for Authors

Dear authors,

Your study is well-designed and conducted, and the presentation of data is very nice. However, there is an issue with inappropriate references that requires consideration.

Additionally, I suggest adding a subsection to the Discussion section on the changes needed to improve the situation. These recommendations should be country-specific and focus on countries with unfavorable trends in stroke mortality, particularly Pakistan and Nepal.

Author Response

1. Your study is well-designed and conducted, and the presentation of data is very nice. However, there is an issue with inappropriate references that requires consideration.

Response: Thank you very much for your insightful comments. We have carefully reviewed all the references and, based on the feedback from other reviewers, we have incorporated additional significant related literature to enhance the study's context and relevance. Thank you again for your kind and constructive feedback.

2. Additionally, I suggest adding a subsection to the Discussion section on the changes needed to improve the situation. These recommendations should be country-specific and focus on countries with unfavorable trends in stroke mortality, particularly Pakistan and Nepal.

Response: Thank you very much for your kind and careful review. South Asia faces numerous challenges related to this disease, and these challenges are indeed similar across countries in the region. In response to your comments, I have added a sentence on strategies to address these challenges. South Asian countries are grappling with numerous challenges related to stroke, and there is an urgent need to develop cost-effective strategies and interventions to control the stroke epidemic. For individual countries, understanding the risk factors and developing targeted measures would be essential in formulating effective stroke strategies for the region. The effective diagnosis and control of hypertension, antiplatelet therapy, and controlling smoking (including the use of chewing tobacco) may be among the most critical intervention areas[1]. We have highlighted the changes we made, and you can find our revision on page 8.

Reference

[1] Wasay M, Khatri IA, Kaul S. Stroke in South Asian countries. Nat Rev Neurol 2014; 10(3): 135-43.

Reviewer 2 Report

Comments and Suggestions for Authors

 Temporal mortality trends attributable to stroke in South Asia:  an age-period-cohort analysis

When given the task of reviewing this paper I was hoping to find an answer to a statement I was taught during my training.  The goal INR ‘s should be lower in south Asians with mechanical valves due to increased risk of hemorrhagic stroke.  The below article supports this notion in ethnic groups treated with warfarin for atrial fibrillation.  After reading the paper I was disappointed that I still do not know why this is true.

Shen AY, Yao JF, Brar SS, Jorgensen MB, Chen W. Racial/ethnic differences in the risk of intracranial hemorrhage among patients with atrial fibrillation. J Am Coll Cardiol. 2007 Jul 24;50(4):309-15. doi: 10.1016/j.jacc.2007.01.098. Epub 2007 Jul 6. PMID: 17659197.

Nonwhites with AF were at greater risk for warfarin-related ICH. Blacks, Hispanics, and Asians were at successively greater ICH risk than whites

Stroke classification is difficult so data reporting strokes is also suspect.  The young or hypertensive are expected to have more hemorrhage. The older have more ischemic, atherosclerotic, or embolic.  Some of these strokes have hemorrhagic conversions.  The classification of stroke is difficult for trained neurologist let alone a family doctor with no imaging ability.

Given these limitations the paper is acceptable but may not be very valuable.  It points the direction like a cane for a blind individual. The differences between countries may be reporting culture.  The tables and figures should be reformatted for easier interpretation.  There does not seem to be enough young strokes to support the following statement.

 “an increase was indicated among males aged 15 to 34 years and females aged 15 to 19 years.  I am not sure I could see this in the data”

I am puzzled by figure 2 E that shows in the 1920’s a peak and decline in stroke in the elderly age group.  This suggests a significant environmental change in this time period.  Was there famine, change in eating due to WWI or abrupt population decline due to influenza to explain the peak and valley?

Data extraction should be made in some form that can be correlated to factors that can be altered.  In the US there are more heart attacks than strokes.  In China there are more strokes than heart attacks.  The salt intake and hypertension in China is much greater than in the US.  It is not provable as a cause and effect but suggest that salt moderation in China should be tested.

Author Response

When given the task of reviewing this paper I was hoping to find an answer to a statement I was taught during my training. The goal INR ‘s should be lower in south Asians with mechanical valves due to increased risk of hemorrhagic stroke. The below article supports this notion in ethnic groups treated with warfarin for atrial fibrillation. After reading the paper I was disappointed that I still do not know why this is true.

Shen AY, Yao JF, Brar SS, Jorgensen MB, Chen W. Racial/ethnic differences in the risk of intracranial hemorrhage among patients with atrial fibrillation. J Am Coll Cardiol. 2007 Jul 24;50(4):309-15. doi: 10.1016/j.jacc.2007.01.098. Epub 2007 Jul 6. PMID: 17659197. Nonwhites with AF were at greater risk for warfarin-related ICH. Blacks, Hispanics, and Asians were at successively greater ICH risk than whites.

Response: Thank you very much for your kind review and assistance in improving this study. Following the reviewer's recommendation, we have carefully reviewed the paper suggested by the reviewer. Atrial fibrillation (AF) is highly prevalent and is associated with a 5-fold increased risk of stroke. Although treatment with warfarin reduces all strokes by 68%, it also increases the incidence of intracranial hemorrhage (ICH) by 0.2% per year, particularly for nonwhites[1]. The recommended paper indicated that nonwhites with AF are at a greater risk for warfarin-related ICH, and the mortality from stroke is higher among South Asians compared to European whites. Cultural and socioeconomic factors may account for part of this excess[2]. To enrich the context of this manuscript, we have added more information about the higher risk of stroke mortality in South Asians in the introduction section and have included the related references. Thank you very much for your helpful comment. You can find our revisions in the introduction section. We have highlighted the changes in yellow.

Reference

[1]Shen AY, Yao JF, Brar SS, Jorgensen MB, Chen W. Racial/ethnic differences in the risk of intracranial hemorrhage among patients with atrial fibrillation. J Am Coll Cardiol. 2007 Jul 24;50(4):309-15.

[2]Gunarathne A, Patel JV, Gammon B, Gill PS, Hughes EA, Lip GY. Ischemic stroke in South Asians: a review of the epidemiology, pathophysiology, and ethnicity-related clinical features. Stroke. 2009 Jun;40(6):e415-23.

Stroke classification is difficult so data reporting strokes is also suspect. The young or hypertensive are expected to have more hemorrhage. The older have more ischemic, atherosclerotic, or embolic. Some of these strokes have hemorrhagic conversions. The classification of stroke is difficult for trained neurologist let alone a family doctor with no imaging ability. Given these limitations the paper is acceptable but may not be very valuable. It points the direction like a cane for a blind individual. The differences between countries may be reporting culture.

Response: Thank you very much for your insightful comment, which has helped to improve this study. We completely agree with the reviewer's observations. In this study, we utilized the World Health Organization (WHO) definition of stroke, which describes stroke as "rapidly developing clinical signs of focal (or global) disturbance of cerebral function, lasting more than 24 hours or leading to death." According to the International Classification of Diseases (ICD), stroke is defined by ICD-10 codes G45–G46.8, I60–I63.9, I65–I66.9, I67.0–I67.3, I67.5–I67.6, I68.1, I68.2, I69.0–I69.3; and ICD-9 codes 430–435.9, 437.0–437.2, 437.5–437.8. In the original data sources, various methodologies have been employed to assess the value of stroke burden, including addressing issues of incompleteness, under-reporting, and misclassification. Corrections have been made, and the redistribution of the garbage codes has been utilized to enhance data quality and comparability. While it is fair to say that the bias in the present study has been significantly reduced, compared to research using raw data without these correction and adjustment steps, we acknowledge that some bias may still exist. We have highlighted this limitation, and you can find it in the discussion section, where we have marked the changes in yellow.

The tables and figures should be reformatted for easier interpretation.

Response: Thank you very much for your comments. In response to the reviewer's feedback, we have made several revisions to the manuscript. Specifically, we have reshaped all the figures to enhance readability. We have increased the text size for axes, titles, and legends, and we have also enlarged the figures within the manuscript. To further strengthen our presentation, we have added a table in the appendix that provides a detailed breakdown of the age-period-cohort (APC) effect. This addition is intended to offer a more robust understanding of the data. Please find our revisions in the updated manuscript.

There does not seem to be enough young strokes to support the following statement. “an increase was indicated among males aged 15 to 34 years and females aged 15 to 19 years. I am not sure I could see this in the data”

Response: Thank you very much for your insightful comments. The local drift can indeed reflect this phenomenon. We observed that the local drift in Pakistan is higher than 0 for the age group of 15 to 34 years, which is not similar to other countries. As recommended by the reviewer, we have added the specific values of the local drift in the appendix file. This addition is intended to help better illustrate this trend.

I am puzzled by figure 2 E that shows in the 1920’s a peak and decline in stroke in the elderly age group. This suggests a significant environmental change in this time period. Was there famine, change in eating due to WWI or abrupt population decline due to influenza to explain the peak and valley?

Response: Thank you very much for your comment. Figure 2 illustrates the decomposition trends of long-term stroke mortality in Bangladesh. Discussing the decline in stroke mortality among older age groups is complex, as it necessitates looking back a considerable time—a century ago—and identifying possible clues. This task is complicated by the fact that Bangladesh gained independence from Pakistan in 1971. We have carefully reviewed the historical context that describes the health status in Bengal. The decrease in stroke mortality may be attributed to transformations in its healthcare system. Faced with the threat of diseases such as cholera, the colonial government began to implement stricter public health measures in Bangladesh and other regions, including improvements in drinking water supply, sewage treatment, and waste management. More hospitals and medical facilities were established in Bangladesh to meet the growing demand for healthcare. These facilities not only served Europeans but also began to provide medical services for local residents. Additionally, some medical research institutions began to emerge in the Bengal region at the beginning of the 20th century[1]. The improvement in health status is also reflected in the continuous increase in life expectancy since the 1920 century[2].

Reference

[1] Pati, B., & Harrison, M. (Eds.). (2008). The Social History of Health and Medicine in Colonial India (1st ed.). Routledge. https://doi.org/10.4324/9780203886984

[2] Aaron O'Neill. Life expectancy (from birth) in Bangladesh from 1865 to 2020. Available online: https://www.statista.com/statistics/1071009/life-expectancy-bangladesh-historical/ (accessed on July 17, 2024).

Data extraction should be made in some form that can be correlated to factors that can be altered. In the US there are more heart attacks than strokes. In China there are more strokes than heart attacks. The salt intake and hypertension in China is much greater than in the US. It is not provable as a cause and effect but suggest that salt moderation in China should be tested.

Response: Thank you very much for your insightful and profound comments. This study describes and compares the long-term trends of stroke in South Asia and detects the underlying effects within these trends. The findings of this study may provide more useful information to understand the stroke burden in the general population. It would indeed be beneficial to explore the correlation between risk factors and stroke, which could offer more valuable insights to address this public health issue. However, due to the limited length of this manuscript, we cannot simultaneously describe this correlation. This constraint is compounded by the fact that there are 23 risk factors in the original data resources from the Global Burden of Disease (GBD). We greatly appreciate the reviewer's kind review and comments. To address this limitation, we have added more information in the limitations section. Kindly find our revision in the discussion section; we have highlighted the changes in yellow.

Reviewer 3 Report

Comments and Suggestions for Authors

1. The manuscript needs to add on proper introduction and information about stroke, at the most adding some basic definition (in terms of SBP) and clinical features about stroke, the diagnosis, the WHO clinical criteria, classification (pathological subtypes), intervention etc., might definitely improvise the overall credibility of the paper.

 2. Similarly, the data sources as discussed in the materials and methods section can be made more elaborative. Especially, the authors can add on details or information about the “Population Attributable Fraction (PAF)” since this is an epidemiological study focused on to the impact of exposures in public health, this will be very informative for the readers.

 3. All the figures included in the manuscript must definitely be formatted for easy readability and clear understanding. Especially the font size of all the images, including the x and y-axis title, labels, legend or key, the data points, etc., must be increased and improvised. Otherwise, it is very difficult to interpret the data that is discussed.

 4. Figure 1. image B; the legend or key denotes the 4 south Asian countries like, Bangladesh (red), Nepal (purple), India (yellow) and Pakistan (green). However, there is an additional key (dotted) denoting the data for south Asia as given in the key along with the above four south Asian countries. I am wondering what are the authors trying to convey.

 5. For an easy interpretation of the data for the readers, I am wondering if it would be possible for the authors to represent the trends in the ASMR for subtypes of stroke (with reference to high SBP) by gender with the consecutive years for each country separately denoting the percentage in the with the individual data point?

6. Similarly, the authors have to be consistent in the format of the data representation in all the figures. The figure 2, the figure axis title, labels, legend or key, the data points, etc., must be improvised. Especially the key or the legend, must be represented for each country separately or have a common key that will be applied to all the countries.

 7. The data representing the longitudinal age curves (figure 4) is not at all readable and very difficult to interpret the data. All the aspects of a legible figure representation has to be strictly followed. I would strongly suggest to re-do this figure. In fact, I would strongly suggest the authors to re-do all the figures to improvise the integrity of this manuscript.

 8. As an add on for the limitations of the study, the authors may have to include somewhere about the occurrence of “ecological fallacy” since the overall study plan has not focused on to the individual level. This has to be discussed as one of the shortcomings of the study for a futuristic troubleshooting.

Comments on the Quality of English Language

The manuscript MUST be thoroughly revised for English language (grammar, syntax and spelling). There are innumerable instances with wrong spellings, wrong grammar, wrong sentence construction, etc. The whole manuscript requires thorough editing.

Author Response

1.The manuscript needs to add on proper introduction and information about stroke, at the most adding some basic definition (in terms of SBP) and clinical features about stroke, the diagnosis, the WHO clinical criteria, classification (pathological subtypes), intervention etc., might definitely improvise the overall credibility of the paper.

Response: Thank you very much for taking the time to help improve this study. We greatly appreciate your effort and assistance. Following the reviewer's recommendation, we have expanded the information on stroke in this manuscript. This includes a comprehensive overview of stroke, its definition, classification, and potential interventions. We have incorporated this information not only in the introduction section but also included the World Health Organization's (WHO) definition of stroke in the methods section. Kindly find our revisions in the methods section. We have highlighted the additions in yellow for easy identification.

2.Similarly, the data sources as discussed in the materials and methods section can be made more elaborative. Especially, the authors can add on details or information about the “Population Attributable Fraction (PAF)” since this is an epidemiological study focused on to the impact of exposures in public health, this will be very informative for the readers.

Response: Thank you very much for your kind comments. In this study, the data sources were extracted from the Global Burden of Disease (GBD) study. In the GBD, stroke mortality estimation is based on verbal autopsy and vital registration data. A standard CODEm approach was employed to model deaths from stroke. CODEm is a flexible modeling tool that utilizes geospatial relationships and information from covariates to produce estimates of mortality for all locations across the time series from 1990 to 2019[1]. Although the PAF (Population Attributable Fraction) value was not used in the estimation of fatal outcomes of stroke, PAF was used to estimate the risk factors for stroke[2]. In response to the reviewer's commendation, we have added more detailed information in the methods section. Kindly find our revision in the text highlighted in yellow.

Referecne:

[1] GBD 2019 Diseases and Injuries Collaborators. Global burden of 369 diseases and injuries in 204 countries and territories, 1990-2019: a systematic analysis for the Global Burden of Disease Study 2019. Lancet 2020;396(10258):1204-22.

[2] GBD 2019 Stroke Collaborators. Global, regional, and national burden of stroke and its risk factors, 1990-2019: a systematic analysis for the Global Burden of Disease Study 2019. Lancet Neurol 2021, 20, 795-820, doi:10.1016/s1474-4422(21)00252-0.

3.All the figures included in the manuscript must definitely be formatted for easy readability and clear understanding. Especially the font size of all the images, including the x and y-axis title, labels, legend or key, the data points, etc., must be increased and improvised. Otherwise, it is very difficult to interpret the data that is discussed.

Response: Thank you very much for your comments. As recommended by the reviewer, we have carefully rechecked all the figures and revised various elements to enhance clarity. These revisions include adjustments to the font size, as well as improvements to the x and y-axis titles, labels, and the legend or key. Thank you for your help in improving this study.

4.Figure 1. image B; the legend or key denotes the 4 south Asian countries like, Bangladesh (red), Nepal (purple), India (yellow) and Pakistan (green). However, there is an additional key (dotted) denoting the data for south Asia as given in the key along with the above four south Asian countries. I am wondering what are the authors trying to convey.

Response: Thank you for your comments. The dotted grey line in Figure 1 indicates the overall Age-standardized mortality rates of stroke in South Asia (Panel A), the total number of deaths resulting from stroke in South Asia (Panel C), and the relative proportion of stroke deaths to all deaths in South Asia. To enhance clarity, we have added more descriptive text to this figure. Kindly find our revised description under Figure 1.

5. For an easy interpretation of the data for the readers, I am wondering if it would be possible for the authors to represent the trends in the ASMR for subtypes of stroke (with reference to high SBP) by gender with the consecutive years for each country separately denoting the percentage in the with the individual data point?

Response: Thank you for your kind comment. In the Global Burden of Disease (GBD) estimation, stroke is categorized into subarachnoid hemorrhage, intracerebral hemorrhage, and ischemic stroke. However, it does not provide subtypes that are directly correlated with systolic blood pressure (SBP). Following the reviewer's commendation, we have added the long-term trends of these three stroke subtypes in the appendix file to provide more detailed information. Thank you again for your kind help.

6. Similarly, the authors have to be consistent in the format of the data representation in all the figures. The figure 2, the figure axis title, labels, legend or key, the data points, etc., must be improvised. Especially the key or the legend, must be represented for each country separately or have a common key that will be applied to all the countries.

Response: Thank you very much for your kind review. As recommended by the reviewer, we have carefully reviewed all the figures and have remade them to ensure consistency throughout the manuscript. To further clarify our presentation, we have also included additional relevant data in a table within the appendix file. Thank you again for your careful review and helpful comments.

7. The data representing the longitudinal age curves (figure 4) is not at all readable and very difficult to interpret the data. All the aspects of a legible figure representation has to be strictly followed. I would strongly suggest to re-do this figure. In fact, I would strongly suggest the authors to re-do all the figures to improvise the integrity of this manuscript.

Response: Thank you for your careful review. In response to the reviewer's comments, we have remade all the figures. We have enlarged the label text, enhanced the definition, and made other revisions to improve clarity and readability. Please see our revisions in the main text.

8. As an add on for the limitations of the study, the authors may have to include somewhere about the occurrence of “ecological fallacy” since the overall study plan has not focused on to the individual level. This has to be discussed as one of the shortcomings of the study for a futuristic troubleshooting.

Response: Thank you so much for your comments. We acknowledge that this study has limitations. It is indeed an ecological study, and there is an inherent risk of the ecological fallacy. Interpretations from results at the population level do not necessarily apply to individuals. We appreciate your thoughtful comment. As recommended by the reviewer, we have added a sentence to declare this limitation. Kindly find our revision in the limitations section.

Round 2

Reviewer 3 Report

Comments and Suggestions for Authors

Thank you for responding to my comments and suggestions.

However, please address the following:

1. I notice minor corrections and amendments to the introduction sectiont from the authors’ side. However, the introduction section still seems to be lacking substance. For example, there is an immense necessity to define and describe about the pathophysiology of stroke in general, which is still missing in the edited draft. The authors need to describe the overall generalized categories of stroke, viz., the Ischemic and Hemorrhagic, which apparently is sub classified as  Intracerebral and subarachnoid (hemorrhagic), the TOAST classification (cardioembolism, small vessel occlusion, large artery atherosclerosis, stroke of undetermined etiology, etc.,)

The introduction can definitely improvised furthermore to at least briefly signifying the above points. Adding two or three lines may not be suffice, please improvise the information. Besides, please thoroughly check the spelling (“stroke” is misspelt as “stoke”) and syntax of the highlighted amendment.  

 2. Similarly, the rebuttal for “Population Attributable Fraction (PAF)”, I understand the authors have followed the methodology or strategy from the GBD 2019 flagship paper, however, it would be sensible to briefly describe the overall selection criteria, and the validation as to why to choose it?

3. Lastly, the manuscript MUST be thoroughly revised for English language (grammar, syntax and spelling). There are innumerable instances with wrong spellings, wrong grammar, wrong sentence construction, etc. The whole manuscript requires thorough editing.

Comments on the Quality of English Language

The manuscript MUST be thoroughly revised for English language (grammar, syntax and spelling). There are innumerable instances with wrong spellings, wrong grammar, wrong sentence construction, etc.

The whole manuscript requires thorough editing.

Author Response

  1. I notice minor corrections and amendments to the introduction sectiont from the authors’ side. However, the introduction section still seems to be lacking substance. For example, there is an immense necessity to define and describe the pathophysiology of stroke in general, which is still missing in the edited draft. The authors need to describe the overall generalized categories of stroke, viz., the Ischemic and Hemorrhagic, which apparently is sub classified as  Intracerebral and subarachnoid (hemorrhagic), the TOAST classification (cardioembolism, small vessel occlusion, large artery atherosclerosis, stroke of undetermined etiology, etc.) The introduction can definitely improvised furthermore to at least briefly signifying the above points. Adding two or three lines may not be sufficient; please improvise the information. In addition, please thoroughly check the spelling (“stroke” is misspelt as “stoke”) and syntax of the highlighted amendment.

Response: Thank you very much for your insightful comments to help improve this study. In accordance with the reviewer’s comments, we added more descriptions of stroke, including the pathophysiology of stroke, generalized categories of stroke, and more important background information. Additionally, we carefully rechecked the word spelling in this section and revised all the spelling mistakes. To guarantee the quality of this study, we also apply the grammar and spelling polishing from MDPI language edit servers in the next process, which provides basic help to address this limitation. Please see our revision in the manuscript; we have yellowed what we changed.

  1. Similarly, the rebuttal for “Population Attributable Fraction (PAF)”, I understand the authors have followed the methodology or strategy from the GBD 2019 flagship paper, however, it would be sensible to briefly describe the overall selection criteria, and the validation as to why to choose it?

Response: Thank you for the comments. In response to the reviewer’s comments, we briefly described the overall selection criteria and the validation as to why to choose it. In the GBD 2019, population attributable fractions (PAFs) were used to analyze the attributable burden of stroke on disability-adjusted life-years. The relative risks in assessing PAF were generated from meta-analyses of epidemiological studies [1]. Additionally, to make this clearer, we added a brief process to estimate mortality attributed to stroke; please see our revision in the revised manuscript.

Reference

GBD 2019 Diseases and Injuries Collaborators. Global burden of 369 diseases and injuries in 204 countries and territories, 1990--2019: a systematic analysis for the Global Burden of Disease Study 2019. Lancet 2020;396(10258):1204-22. doi: 10.1016/s0140-6736(20)30925-9 [published Online First: 2020/10/19]

  1. Last, the manuscript MUST be thoroughly revised for English language(grammar, syntax and spelling). There are innumerable instances with wrong spellings, wrong grammar, wrong sentence construction, etc. The whole manuscript requires thorough editing.

Response: Thank you very much for your kind review. As recommended by the reviewer, we have rechecked the manuscript, including grammar and spelling. Additionally, we will also apply for language editing services from the MDPI language edit servers to help polish this manuscript in the next process. Thank you again for your careful review.